# Tailored Biobased Resins from Acrylated Vegetable Oils for Application in Wood Coatings

Sabine Briede [1], Oskars Platnieks [1,*], Anda Barkane [1], Igors Sivacovs [2], Armands Leitans [3], Janis Lungevics [3] and Sergejs Gaidukovs [1,*]

[1] Institute of Polymer Materials, Faculty of Materials Science and Applied Chemistry, Riga Technical University, P. Valdena 3/7, LV-1048 Riga, Latvia; sabine.briede@rtu.lv (S.B.); anda.barkane@rtu.lv (A.B.)
[2] JSC Olaine Chemical Plant BIOLAR, Rupnicu 3, LV-2114 Olaine, Latvia; igors.sivacovs@biolar.lv
[3] Department of Mechanical Engineering and Mechatronics, Faculty of Mechanical Engineering, Transport and Aeronautics, Riga Technical University, Kipsalas 6B, LV-1048 Riga, Latvia; armands.leitans@rtu.lv (A.L.); janis.lungevics@rtu.lv (J.L.)
* Correspondence: oskars.platnieks_1@rtu.lv (O.P.); sergejs.gaidukovs@rtu.lv (S.G.)

**Abstract:** The modern coating market is dominated by acrylic, polyurethane, and polyester polymer resins produced from unsustainable fossil resources. Herein, we propose the preparation of resins from biobased components to produce functional and solvent-free wood coatings with enhanced performance properties. Acrylated rapeseed, linseed, and grapeseed oils were prepared via a one-step synthesis and used as a basis for the control of resin viscosity and fatty acid content. A combination of vegetable oil acrylates was used as a matrix and the biobased monomer propoxylated glycerol triacrylate (GPT) was selected to tailor the properties of the UV crosslinked network. During polymerization, the GPT monomer induced a two-phase microstructure as indicated by an SEM analysis. The possibility of generating a tailored microstructure in the final material was examined in this study. The addition of GPT increased the storage modulus by up to five-fold, crosslink density by up to two-fold at 20 °C, and glass transition temperature by up to 10.2 °C. Pull-off adhesion tests showed a strength of 1.21 MPa. In addition, the photo-oxidation effect on samples, i.e., aging, was assessed with microhardness, sliding friction, and optical microscopy. Coatings showed a microhardness value up to 250 MPa, while a coefficient of friction ($\mu$) was in the range of 0.21 to 0.88.

**Keywords:** UV photopolymerization; photo-oxidative degradation; microhardness; adhesion; sliding friction

## 1. Introduction

Coatings contribute significantly to material protection and preservation. With the current global sustainability issues, wood is the material of choice for modern engineering solutions and functional and decorative applications. Recent trends show that sustainability is being incorporated into the coatings industry [1]. So far, the sustainability approach has targeted reduced energy consumption by utilizing ultraviolet (UV)-curable coatings that use high solid content or solvent-free technology. The UV approach has proven to be more efficient than thermal curing [2]. The next step is switching to UV-curable monomers and oligomers sourced from renewable resources. This research undertakes the challenge of formulating biobased coating resins by incorporating the energy-saving aspects of UV-curing technology.

Vegetable oils are characterized by biodegradability, low toxicity, high flash points (>300 °C), and low flammability and are considered sustainable [3]. These excellent properties play a role in their selection as wood-coating materials. However, some drawbacks to vegetable oils (VOs) and their derivatives (susceptibility to oxidative polymerization reactions, low mechanical strength, and low toughness due to long aliphatic chains) [4] must be addressed. Nevertheless, these properties can be improved by adding additives, selecting

a modification route for vegetable oils, and controlling the fatty acid composition [5]. In our previous research, we demonstrated that one-step acrylation is suitable for preparing relatively low-viscosity resin for UV-assisted 3D printing [6]. Therefore, the modification route of choice was the optimized acrylation route for coating preparation.

VO has been demonstrated as a perspective source for UV-curable coatings, including linseed [7], castor [8], tung [9], jatropha, and palm oils [10]. Likewise, various properties are intentionally designed from oil derivatives. The production, quality, and processing of polymer coatings depend highly on rheological behavior [11]. UV-curable coatings are characterized mainly by high viscosity, which is one of the major challenges in broadening their application. Therefore, non-reactive or, most often, reactive diluents, i.e., monomers, are used to reduce viscosity and improve fluidity [12,13]. Relatively low-viscosity soybean oil acrylate (5406 mPa·s) was synthesized using 2-hydroxyethyl acrylate as an acrylic agent to obtain acrylic ethers instead of esters [14]. The authors concluded that, compared to ester linkages, ethers have a greater free volume due to their freedom of rotation, meaning lower viscosity can be obtained. Hongjie et al. created another biobased acrylate by reacting soybean oil polyol with acryloyl chloride at low temperatures and without using a catalyst or inhibitor [15]. By changing the molar ratio, the obtained products had a lower viscosity of 208 and 192 mPa·s compared to acrylated epoxidized soybean oil (AESO) with a reported viscosity of 1225 mPa·s. The authors observed that introducing unsaturated bonds reduced the product's viscosity. Fei et al. reported the addition of three reactive diluents for AESO: styrene, 1,4-butanediol dimethacrylate (BDDMA), and trimethylolpropane trimethacrylate (TMPTMA) [16]. The viscosity for these resins decreased from 3352 mPa·s for pure AESO to 44, 165, and 624 mPa·s for the AESO/diluent system at 30 °C, respectively. Thus, according to the literature, frequently used reactive diluents for the acrylated vegetable oil (AVO) matrix are primarily derived from petroleum feedstocks and are often volatile and harmful. Some studies reported using AVOs as reactive diluents [17], or other renewable feedstocks such as eugenol [18], methacrylated isosorbide [19], etc. Dai et al. developed a high-performance UV-curable coating from AESO, fossil-based additives, and the crosslinking agent IG synthesized from itaconic acid and glycidyl methacrylate with a biobased content of 31.4% [20]. The rigid coating showed an increase in mechanical strength from 1.6 MPa (pure AESO) to 6.0 and 10.1 MPa (20 and 40 wt% IG added), while the tensile modulus rose significantly by 670% and 1932%, respectively. Despite all the success of reducing the resin viscosity, the lack of commercially available biobased coatings shows that more research is still required. Emphasis should also be placed on biobased solutions' mechanical and aging performance.

In our work, phase separation was induced as a strategy to create an enhanced microstructure. The stability and low viscosity of the initial homogeneous solution are benefits of the phase separation process in terms of toughening thermosetting networks [21]. The possibility of generating a tailored microstructure in the final material was examined in this study. Distinct polymeric microstructures were produced by the addition of 5 and 20 wt% of propoxylated glycerol triacrylate (GPT) in the AVO matrix and studied by SEM. This work expanded on the plant oil derivatives by introducing acrylated rapeseed, linseed, and grapeseed oils as value-added monomers. The resin formulations focused on the use of biobased components. Furthermore, extensive mechanical and photo-oxidation (aging) studies explored the feasibility of actual biobased wood coatings, not just the concept.

## 2. Materials and Methods

### 2.1. Materials

Unrefined rapeseed and linseed oils were purchased from a local producer "Iecavnieks" (Latvia) and the grapeseed oil used in this study was "Monini" (Italy). The synthesis of acrylated rapeseed, linseed, and grapeseed oil (R, L, and G, respectively) was described in our previous work [6]. Briefly, the selected oil, acrylic acid, and $BF_3 \cdot OEt_2$ were mixed in a round-bottom flask and stirred for 5 h at 80 °C before being left overnight at room temperature (22 ± 1 °C). In the work-up, the organic phase was dissolved in hexane, washed with

the NaHCO$_3$ solution, dried, filtered, and evaporated under reduced pressure. BF$_3$·OEt$_2$, hexane, NaHCO$_3$, and photoinitiator diphenyl(2,4,6-trimethylbenzoyl)phosphine oxide (TPO) were supplied by Sigma Aldrich. In addition, propoxylated glycerol triacrylate (GPT) was added as a reactive diluent and was supplied by Arkema. The biobased content of GPT was 15%. The chemical structures of the investigated components and their application are illustrated in Figure 1.

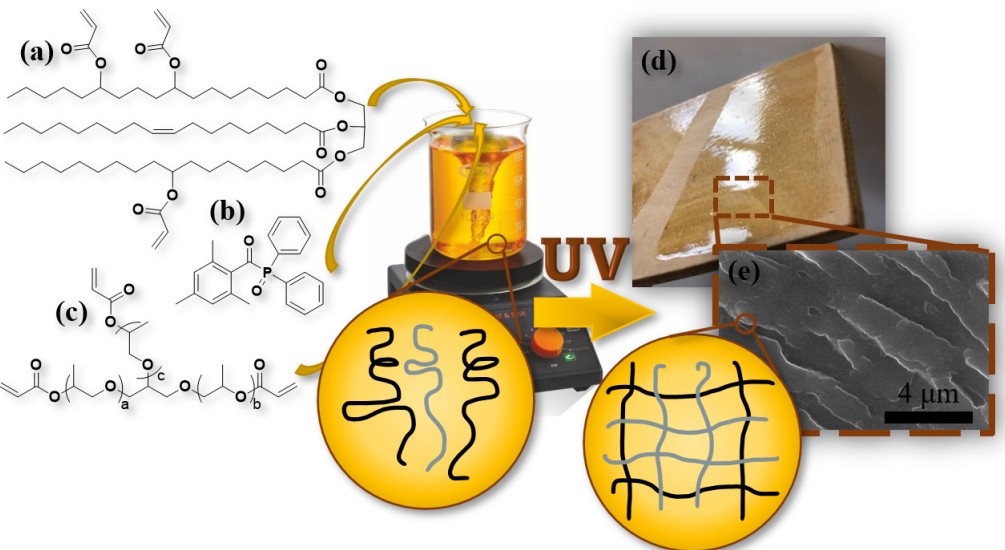

**Figure 1.** Scheme for this study: (**a**) acrylated vegetable oil (generalized structure); (**b**) TPO photoinitiator; (**c**) GPT monomer; (**d**) ultraviolet (UV)-cured acrylated vegetable oil coating on plywood; (**e**) formation of microstructure, SEM image.

### 2.2. UV-Curing Process of Coatings

Acrylated R, L, and G oils—a series of yellowish oily products—were mixed in a mass ratio of 50/50 to obtain a viscosity that was approximately the same for all compositions, as well as fatty acid residues in a similar ratio. The representative fatty acid residues in R were oleic acid (~58.8%), L—linolenic acid (~48.8%), and G—linoleic acid (~74.7%) [22–24]. GPT was used as an additional monomer in the UV-curable system and acted as a reactive diluent. In addition, GPT can considerably enhance mechanical strength since the trifunctional monomer leads to higher crosslinking density [25]. Nine resin compositions were prepared to study acrylated vegetable oils' (AVOs') application for wood coatings: three acrylated vegetable oil resins, three resins with 5 wt% GPT (named G5), and three with 20 wt% GPT (named G20) (Table 1). At the same time, 3 wt% of the TPO photoinitiator was dissolved in less than one milliliter of acetone and added to the compositions of acrylated oils. Finally, liquid coatings were applied on 130 mm × 130 mm × 6.2 mm 5-layer birch plywood substrate using an applicator with a wet film thickness of 5 mils (127 microns). One coating specimen occupied an area of approximately 100 cm$^2$ of plywood. Samples were then cured under monochromatic conditions ($\lambda$ = 405 nm) at a UV intensity of 0.13 μW/cm$^2$ at a distance of 3 cm. The optimal curing, i.e., crosslinking, time of coatings was 70 s, determined by FT-IR spectroscopy, as described in our previous work [6]. The conversion of vegetable oil acrylic double bonds ranged from 70 to 90%, which can be seen in Supplementary Materials Figure S1. Acrylates subjected to radical polymerization are vulnerable to inhibition by molecular oxygen, leading to incomplete curing, particularly manifested by a tacky upper surface or even complete non-curing [26]. For this reason, samples were washed with acetone to remove the unreacted resin. In addition, free-standing coating films were obtained by applying resin on the glass substrate before they were peeled off. Properties such as storage modulus and tensile strength were measured from the prepared free-standing films.

**Table 1.** Resin composition of biobased acrylates for wood coating application.

| Biobased Resins | Resin Composition | Monomers (wt%) | | | |
|---|---|---|---|---|---|
| | | **R** | **L** | **G** | **GPT** |
| RL | R/L (50/50) | 50 | 50 | 0 | 0 |
| RG | R/G (50/50) | 50 | 0 | 50 | 0 |
| LG | L/G (50/50) | 0 | 50 | 50 | 0 |
| RL_G5 | R/L (50/50) 95 wt%, GPT 5 wt% | 47.5 | 47.5 | 0 | 5 |
| RG_G5 | R/G (50/50) 95 wt%, GPT 5 wt% | 47.5 | 0 | 47.5 | 5 |
| LG_G5 | L/G (50/50) 95 wt%, GPT 5 wt% | 0 | 47.5 | 47.5 | 5 |
| RL_G20 | R/L (50/50) 80 wt%, GPT 20 wt% | 40 | 40 | 0 | 20 |
| RG_G20 | R/G (50/50) 80 wt%, GPT 20 wt% | 40 | 0 | 40 | 20 |
| LG_G20 | L/G (50/50) 80 wt%, GPT 20 wt% | 0 | 40 | 40 | 20 |

*2.3. Viscosity*

The viscosity of liquid coating resin was measured using an MCR102 rheometer from Anton Paar (Graz, Austria). The instrument was equipped with a 25 mm diameter spindle with plate-plate geometry, and the measurement gap was set to 0.1 mm. The changes in the viscosities with temperature were calculated using the Arrhenius relationship:

$$\eta = \eta_0 e^{\frac{E_a}{RT}} \tag{1}$$

where $\eta$ (Pa·s)—viscosity, $\eta_0$—pre-factor, $E_a$ (J/mol)—activation energy for the viscous flow, $R$ = 8.31 (J/K·mol)—ideal gas constant, and $T$ (K)—absolute temperature.

*2.4. Scanning Electron Microscopy*

Scanning electron microscopy (SEM) was used for the surface morphology studies of coated plywood using a NovaNano SEM 650 (Hillsboro, OR, USA). The fracture surface was prepared with liquid nitrogen, and images were taken of the cross-sections of the coatings. The samples were attached to standard aluminum pin stubs with electrically conductive double-sided carbon tape, and the image was generated with 10 kV acceleration voltage coatings.

*2.5. Dynamic Mechanical Analysis*

The storage modulus, loss modulus, and loss factor of the cured free-standing coating films were recorded using a dynamic mechanical analysis (DMA) Mettler DMA/SDTA861e (Mettler Toledo, Columbus, OH, USA) analyzer. An experiment was carried out in tension mode from −70 to 100 °C with a 3 °C/min heating rate, the applied force of 10 N and 10 μm elongation, and a frequency of 1 Hz. The glass transition temperature ($T_g$) was defined by the maximum of the loss modulus curve. The photopolymerized coating films were prepared with dimensions of 20 mm × 4 mm and a thickness of approximately 0.2 mm. The crosslink density ($\nu_e$) was calculated according to an equation from the literature [27]:

$$\nu_e = \frac{E'}{3RT'} \tag{2}$$

where $\nu_e$ (mol/m$^3$)—molar concentration of crosslinks, $E'$ (Pa)—represents the storage modulus of the samples at the rubbery plateau, $R$ = 8.31 (J/K·mol)—universal gas constant, and $T'$ (K)—temperature corresponding to the storage modulus value.

*2.6. Tensile Tests*

Tensile tests were performed on cured free-standing coating films with dimensions of 40 × 10 and a thickness of approximately 0.2 mm to study the tensile behavior of coatings. Elongation at break, tensile strength, and Young's modulus were recorded on Tinius Olsen

model 25ST (Horsham, PA, USA). All tensile tests were conducted at a strain rate of 1 mm/min and were repeated at least five times. The average results were reported.

### 2.7. Pull-Off Adhesion Test

Pull-off adhesion tests were adopted to observe the interfacial bonding between AVO coatings and birch plywood substrate based on modified LVS EN ISO 4624:2003. First, the AVO coatings were applied on the plywood substrate using an applicator before it was treated with sandpaper (medium grit), providing a smooth plywood surface. The coatings were then cured under UV light. Next, 20 mm metal dollies were sanded with 80- and 180-grit sandpapers and cleaned with isopropanol. Then, 2K Bison epoxy resin glue was used as an adhesive. Dollies were glued to the coating surface at least 25 mm apart, and the adhesive was allowed to cure for 24 h. After that, the cut along the circumferences of the dollies was performed and the dollies were pulled off using a PoliTest AT-M adhesion tester at 0.25 MPa/s load intensity. The measurements were repeated six times, and the average result was reported. After the removal of the dolly, a visual examination of the surface was performed to determine the nature of joint failure (cohesive, adhesive, or mixed).

### 2.8. Photo-Oxidation

Photo-oxidation of the coated plywood was carried out under an Hg-UV lamp with an intensity of 9400 mW/cm$^2$, at a distance of 30 cm.

### 2.9. Microhardness

Microhardness testing of photo-oxidized wood coatings was performed with Vickers, Vickers Instruments (York, UK). The Vickers microhardness (in MPa) was determined according to the equation:

$$Microhardness = \frac{2 \cdot P \cdot 9.807 \cdot \sin\left(\frac{\alpha}{2}\right)}{\left(d \cdot k \cdot \frac{n}{1000}\right)^2} = 1.854 \cdot \frac{P \cdot 9.807}{\left(d \cdot k \cdot \frac{n}{1000}\right)^2} \tag{3}$$

where *P*—load (kg); *α*—angle between opposite faces of the diamond = 136°; *n*—interval; *d*—mean diagonal of the indentation (mm); *k*—equipment coefficient that depends on applied load.

The typical Vickers diamond pyramid was used as an indenter (apex angle of 136°), which was forced onto the surface of the specimens. Then, the 0.2 kg load was selected with an indentation time of 30 s. The area of the remaining indentation after the retraction of the diamond pyramid was calculated from the remaining imprint's diagonals visible in the microscope. Indentation diagonals were measured under a magnification of 4×. The test was repeated three times, and an average result was reported.

### 2.10. Sliding Friction Test

Sliding friction measurements were performed on coated plywood using a pin-on-disc tribometer *TRB*$^3$ (CSM Instruments, Peseux, CH) under dry friction conditions according to ASTM G99-05 ("Standard test method for wear testing with a pin-on-disk machine"). Investigated coatings were tested against a standardized 100Cr6 (EN 683-17 "Heat-treated steels, alloy steels and free-cutting steels—Part 17: Ball and roller bearing steels") steel ball (Ø 6 mm) as a stationary counter body or triboindenter. All the tribotests were performed for a 25 m sliding distance applying the indenter load of 0.5, 1.5, and 3.0 N; the tribotrack radius was set to 3 mm, and there was a linear sliding speed of 0.03 m/s. All tests took place at room temperature.

## 3. Results and Discussion

### 3.1. Physical Properties

Viscosity is affected by the fatty acid chain length and degree of unsaturation [28]. The viscosity of liquid vegetable oil-based coating resins decreased with reactive diluents

(Figure 2a). Acrylated vegetable oils (AVOs) were mixed with propoxylated glycerol triacrylate (GPT), which acts as a reactive diluent during resin formulation and becomes an integral part of the coating during curing [29]. The addition of GPT monomer to the AVO reduced the viscosity by an average of 550 mPa·s for all three resins. Viscosity affects molecular mobility when exposed to UV irradiation. Low viscosity affects the diffusion of the reactive species and therefore increases the mobility of the monomers. Zong et al. concluded that relative reactivity is proportional to viscosity [30]. The authors added vinyl ether that reduced the viscosity of the resin and led to higher diffusion rates for the propagating cationic species, resulting in the acceleration of linseed oil derivative polymerization. The increased mobility of the molecules increases the lifespan of free radicals, thus reducing disproportionation and termination reactions [31].

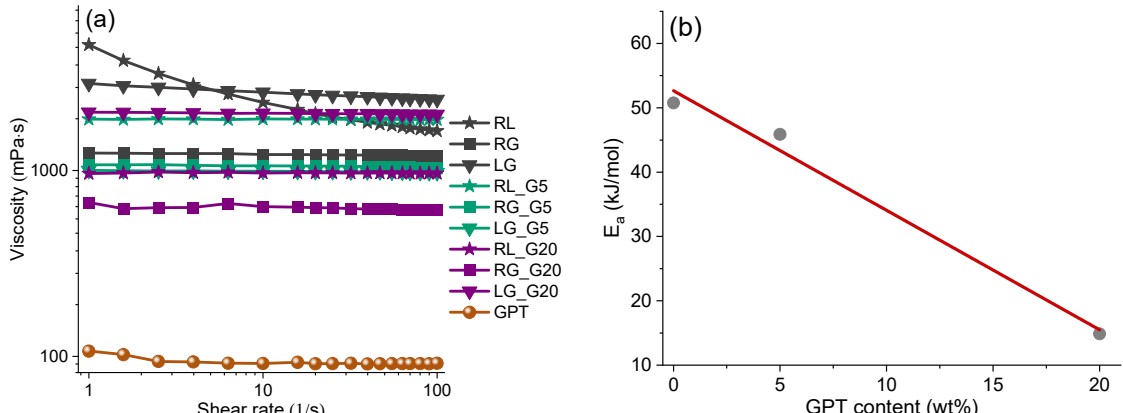

**Figure 2.** (**a**) Viscosity of acrylated vegetable oil resins with GPT; (**b**) activation energy ($E_a$) drop with an increasing GPT content in RG coating.

The Arrhenius relationship (Equation (1)) was used to assess how viscosities changed with temperature. Calculating the pre-factor $\eta_0$ requires extrapolating to infinite temperature; therefore, it is very challenging to measure it accurately. Viscosity decreased exponentially with temperature (Supplementary Materials Figure S2). As a result, a slight increase in the temperature of the acrylated resins can reduce their viscosities significantly, which makes them easier to process. The viscosities of resins that include GPT varied only slightly compared with AVOs; nevertheless, that influenced the activation energy ($E_a$). For RG samples, we observed a linear decrease in $E_a$ with an increase in GPT content (Figure 2b). The GPT molecule is smaller than AVOs' molecules, decreasing and disrupting the intermolecular interactions among the larger AVO monomers [32]. Thus, monomer molecules can slide past one another, reducing the activation energy for viscous flow [33].

AVOs and GPT are mutually soluble when in the form of monomers (Table 2). To determine the polarity of GPT, its solubility was tested in four different solvents with an increasing polarity. As expected, GPT dissolved in ethanol and methanol, confirming the polar characteristic of the molecule. This was also supported by the fact that GPT was insoluble in non-polar rapeseed oil. The polarity of the GPT molecule is most similar to that of alcohol, while water is too polar for GPT.

**Table 2.** Solubility of GPT monomer in different solvents at room temperature.

| | Liquid | GPT Solubility |
|---|---|---|
| | Rapeseed oil | Insoluble |
| | AVOs | Soluble |
| Increased polarity | Ethanol | Soluble |
| | Methanol | Soluble |
| | Water | Insoluble |

### 3.2. SEM Structure Studies

The cross-section images of RL, RL_G5, and RL_G20 coatings were obtained with SEM (Figure 3). SEM revealed the formation of two phases in the microstructure when GPT was added. Induced microstructure directly relates to the separation behavior and local order of reactive groups [34]. Two distinctly different types of microstructures can be observed. GPT as a photocurable polar monomer segregated in domains, creating cavities in the AVO-rich microphase. Likewise, more hydrophobic AVOs will preferentially segregate in the oil-soluble domains because of the existing long alkyl chains [34]. Large local inclusions can be observed in sample RL_G5 (Figure 3a). It is assumed that one phase was formed by the AVO matrix, while the other phase was formed by the AVO regions that were saturated with GPT. This type of microstructure should result in lowered mechanical properties as this type of morphology promotes local stress concentrations. The addition of 20 wt% GPT resulted in a much finer microstructure with the mixed phase of AVOs and GPT dominating, while regions of saturated GPT became almost invisible in the same magnification (Figure 3b). In addition, the cross-section of RL_20 was much rougher than RL_G5 and RL in 10,000× magnification, indicating that a toughened microstructure was formed, which should result in enhanced mechanical properties (Supplementary Materials Figure S3).

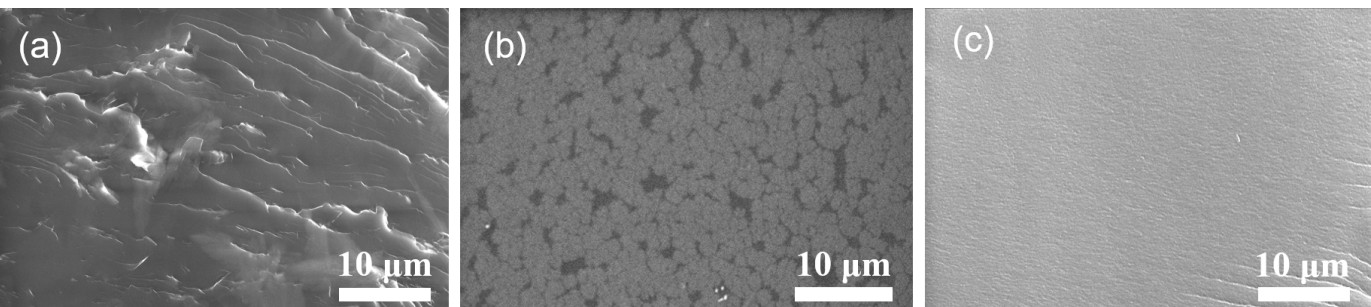

**Figure 3.** Cross-section SEM micrographs of the fracture surface morphology of (**a**) RL, (**b**) RL_G5, and (**c**) RL_G20 in 2500× magnification.

### 3.3. Dynamic Mechanical Properties

A dynamic mechanical analysis (DMA) was conducted on free-standing coating films. The DMA graphs are shown in Figure 4, and some defining characteristics are given in Table 3. Glass transition temperatures ($T_g$) were determined from the maximum of loss modulus (E"), which was described as a more precise method rather than determination from tanδ peaks (Figure 4b) [35]. The storage modulus for all coatings was very high at low temperatures when the material was in a glassy state, especially for coatings with 20 wt% GPT. It decreased rapidly with the temperature rise until the rubbery plateau was reached. As the GPT content increased from 0 wt% to 20 wt%, the E' of coatings increased by 15 MPa at room temperature.

The loss modulus characterizes the dissipated energy and thus signifies the viscous response of the sample. The loss modulus curves reached a maximum with a temperature rise and then decreased due to the maximum energy dissipation brought on by the free movement of the polymeric chains [36]. A correlation between the added GPT content and $T_g$ can be seen. RL_G20, RG_G20, and LG_G20 had $T_g$ 2.7, 10.2, and 5.5 °C higher than their corresponding RL, RG, and LG coatings. $T_g$ shifted to the right due to a decrease in chain mobility and a loss of free volume, which were also the effects of increased crosslinking [37]. However, the $T_g$ values of RL_G5, RG_G5, and LG_G5 were fluctuating due to the local inhomogeneities. The fatty acids in vegetable oils act as plasticizers, allowing more flexibility between the chains and resulting in relatively low $T_g$ values [38]. The fatty acids enlarge the linear chains; therefore, the energy required to cause rotations

around molecular bonds is reduced. As a result, the glass transition temperature shifts to relatively low temperatures [37].

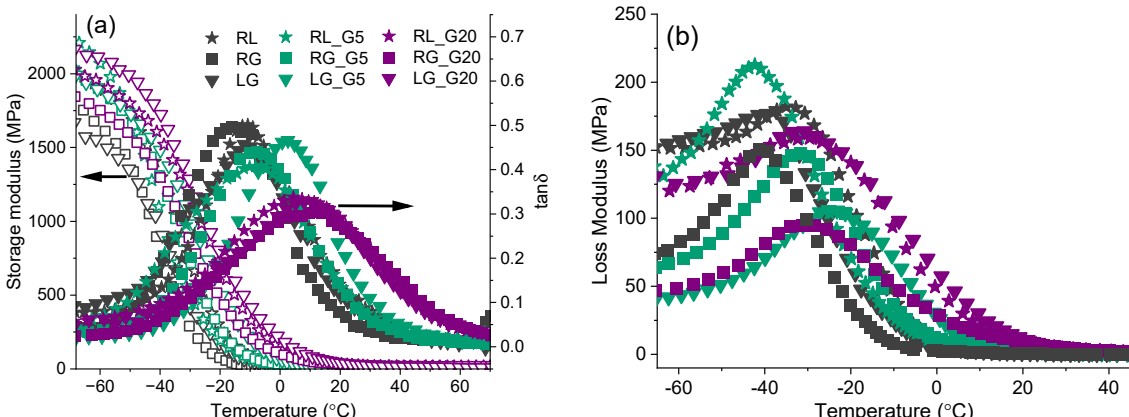

**Figure 4.** DMA plots for photopolymerized AVO coating films: (**a**) storage modulus and tanδ; (**b**) loss modulus.

**Table 3.** Mechanical properties of coatings.

| Coatings | $T_g$ (°C) [1] | $E'$ at Rubbery Plateau (MPa) | $\nu_e$ (mol/m³) | Tensile Strength (MPa) | Elongation at Break (%) |
|---|---|---|---|---|---|
| RL | −34.0 | 13 | 1.5 | 0.70 ± 0.10 | 10.7 ± 1.5 |
| RG | −40.6 | 8 | 1.1 | 0.52 ± 0.08 | 6.3 ± 1.0 |
| LG | −37.6 | 7 | 0.7 | 0.48 ± 0.02 | 8.1 ± 1.1 |
| RL_G5 | −42.3 | 7 | 1.0 | 0.27 ± 0.06 | 2.7 ± 0.5 |
| RG_G5 | −31.2 | 13 | 1.7 | 0.34 ± 0.05 | 3.8 ± 0.4 |
| LG_G5 | −24.7 | 11 | 1.4 | 0.20 ± 0.04 | 2.2 ± 0.2 |
| RL_G20 | −31.3 | 17 | 2.1 | 0.76 ± 0.10 | 5.3 ± 0.6 |
| RG_G20 | −30.4 | 17 | 2.2 | 0.64 ± 0.16 | 3.5 ± 0.1 |
| LG_G20 | −32.1 | 13 | 1.9 | 0.67 ± 0.12 | 4.1 ± 0.1 |

[1] Obtained from loss modulus curve.

As the GPT concentration increased, the tanδ maximum shifted to the right, while the decrease in the height of the tanδ curve for coatings with 20 wt% GPT confirmed the reduced polymer chain mobility and, therefore, higher crosslink density. However, the limited concentration of crosslinked polymers must always be considered. For example, the effect of styrene was examined in soybean oil-based thermosets, where the crosslinking density increased with the styrene content up to 20 wt% but decreased beyond that with an increasing styrene concentration [39]. The change in shape and width of the tanδ peak of the coatings with GPT suggested that the crosslinked network became more complex. Here, the broad peaks indicated the formation of two phases which were prominently seen in compositions with 20 wt% GPT. More time was needed for the relaxation of molecules due to the lower polymeric chain movement resulting from the formation of higher crosslinking densities [36]. Yin et al. compared interpenetrating polymer networks (IPN) prepared from polyurethanes and cottonseed, tung, and castor oil derivatives [40]. Different tanδ peak widths were observed and explained by the competition between two processes in the IPN formation: firstly, polymer chain migration which causes phase separation; and secondly, the interpenetration (chain entanglement) between the two networks, which prevents phase separation. The proposed processes that occur in IPN could apply to the AVO/GPT system, while the presence of the acrylate group in both components suggests that the existence of two fully independent networks is unlikely.

### 3.4. Tensile Properties

The tensile properties of free-standing coating films are shown in Figure 5 and the results are displayed in Table 3. The addition of GPT to AVOs significantly changed the tensile curves (Figure 5a). The addition of 5 wt% GPT yielded overall reduced properties, while samples with 20 wt% GPT increased the stiffness and enhanced the tensile strength of AVOs. Tensile strength depended on crosslink density, the rigidity of the formed crosslinks, the uniformity of the crosslink distribution, and crosslinking agent penetration. Increasing the concentration of GPT in the polymer system provided a higher crosslink density ($\nu_e$ in Table 3) and thus a higher Young's modulus (E) (Figure 5b), which correlates with higher hardness [41]. Coating films with 5 wt% GPT showed lower tensile properties, which could be attributed to the formation of stress concentration promoting local inclusions described in the SEM analysis. A non-uniform crosslinking distribution and boundary regions between phases reduce the tensile strength [42]. In addition, the limited orientation of the long triglyceride molecules can cause high tensile strength loss [43]. The slight increase in Young's modulus for samples with 5 wt% GPT could be explained by the reduced content of oils, which contain saturated fatty acid moieties.

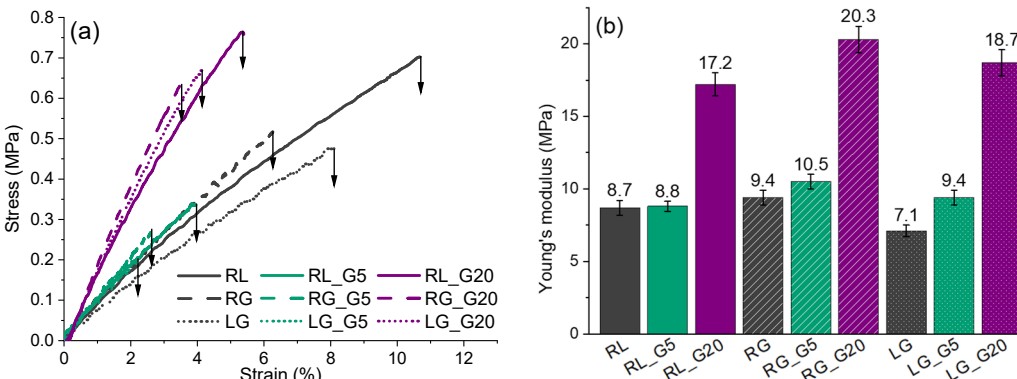

**Figure 5.** Tensile strength of vegetable oil-based coatings; (**a**) stress–strain curves; (**b**) Young's modulus.

Propylene glycol units present in GPT may influence Young's modulus [44]. The addition of GPT increased the modulus up to 20.3 MPa, achieving a 2.0- to 2.6-fold increase compared to coating films without GPT. It can be seen that the most considerable contribution to the elongation of coatings came from linseed oil. This can be explained by the amount of saturated fatty acids in linseed oil (approx. 11.3%). As saturated fatty acids cannot be acrylated, they act as plasticizers [23]. Elongation at break for samples with GPT was below 6%, confirming structural changes induced by the enhanced crosslink density [20].

### 3.5. Adhesion Strength

The coating can only effectively achieve other functions when the interface between it and the substrate is reliable and robust. The pull-off adhesion test was used to evaluate the coating's adhesion to birch plywood. The way the wood substrate is treated and prepared is essential. Surface energy and associated morphology play a crucial role in determining the interfacial parameters, regardless of polymer-coating properties.

The adhesion of coatings presents the combined performance of the two-phase interface between AVO and plywood, which directly affects the lifespan of the substrate. To demonstrate the adhesion strength, we conducted the test only for RL coatings. As shown in Table 4, the force required to remove the AVO coating from the substrate increased with increasing GPT content. The pull-off strength increased from 0.56 MPa for RL to 1.21 MPa for RL_G20. Hence, GPT contributed to a 0.65 MPa increase, which was more than half of the adhesion strength of the RL_G20. In comparison, the interfacial adhesion between

the AESO coating and the starch membrane was reported to be around 2 MPa and was improved after surface treatment with (3-aminopropyl) triethoxysilane (APTES) [45].

**Table 4.** Coatings' pull-off strength with the evaluation of the type of failure.

| Coatings | Average Adhesion Strength (MPa) | Type of Failure |
|---|---|---|
| RL | 0.56 ± 0.12 | Glue failure |
| RL_G5 | 0.87 ± 0.15 | Mixed (adhesive + cohesive failure) |
| RL_G20 | 1.21 ± 0.14 | Mixed (adhesive + cohesive failure) |

After the test, a visual approach was used to determine the type of failure of the coating materials (Figure 6). In both cases with GPT, the failure pattern was mixed (Figure 6b,c). We could observe a failure of the bond between the applied coating and substrate as well as a cohesive failure of the coating itself. In the case of RL (Figure 6a), a glue failure between the applied coating and epoxy resin predominated. In all cases, there were no visible wood fibers remaining on the samples, which means that the substrate failure did not occur. The reason could be the non-sufficient penetrating of resin into the wood substrate, as well as the lack of intramolecular bonds between the coating and wooden substrate. There is potential room for improvement and a future investigation is needed. The RL coating remained tacky, even after the treatment with acetone. The adhesive was practically applied to the surface's unpolymerized part and did not stick properly. Therefore, the coating itself should be able to withstand more stress. This was due to the presence of oxygen as a free radical scavenger during photopolymerization, thus competing with acrylic radicals formed during UV-curing. In some cases, amines are used as oxygen scavengers and function as an efficient sacrificial site for hydrogen and electron abstraction reactions, protecting the material structure [46].

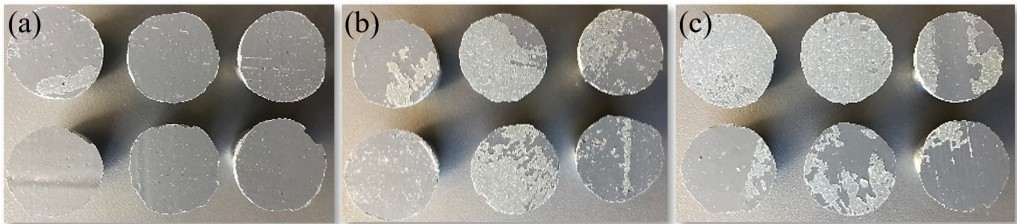

**Figure 6.** Failure patterns of coating materials (**a**) RL; (**b**) RL_G5; (**c**) RL_G20.

*3.6. Photo-Oxidation Effect on Coatings' Properties*

Photo-oxidation of coated plywood was conducted under an Hg-UV lamp for up to 9 h. Microhardness measurements were performed at the surface of photo-oxidized coatings, calculated according to Equation (3), and presented as a function of irradiation time (Figure 7a). The microhardness test was not applicable for AVOs without a GPT monomer because of the observed self-healing effect (the samples were too soft to detect the indentation). Figure 7a reveals a significant increase in microhardness during the irradiation for coatings with added GPT. The microhardness of coatings with 5 wt% of GPT remained constant after about the first hour of exposure, but for coatings with 20 wt% of GPT, the microhardness continued to increase and continuously harden. Local inclusions in coatings with 5 wt% of GPT reflected the relatively lower microhardness, reaching values below 100 MPa. In the case of coatings with 20 wt% GPT, microhardness increased up to 250 MPa, showing the effect of tri-functional acrylic groups on coating strength. During photo-oxidation, crosslinked structures can be produced by any bimolecular combination of radicals. It is essential to know that in the early stages of photo-oxidation, chain scission reactions are less common than oxidative crosslinking coupling. For this reason, there is an increase in the stiffness of the oxidized layer. Dupuis et al. concluded that at short

irradiation times, the crosslinking reactions regulate changes in the top surface's mechanical properties, whereas at longer irradiation times, the β scission reactions become more competitive and reduce the surface stiffness. They measured an increase in crosslink density upon aging by analyzing the $T_g$ of the upper layer of the sample and the bulk material. An increase of 7 °C was detected for the upper layer, while $T_g$ of the bulk material remained unchanged, confirming that new crosslinks were formed mainly in the upper layer of the coating [47].

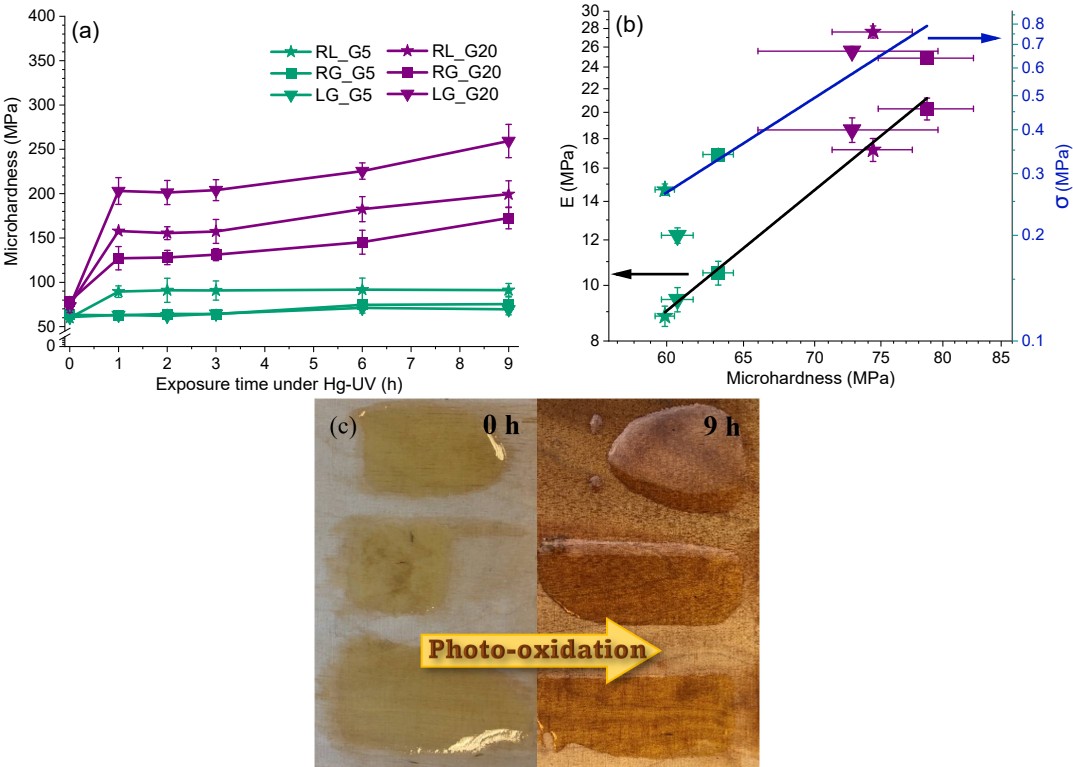

**Figure 7.** (**a**) Change of microhardness of coatings with GPT during Hg-UV exposure; (**b**) linear relationship between microhardness (0 h) and Young's modulus or tensile strength for coatings with GPT (log-log scale); (**c**) plywood and coatings after 9 h of photo-oxidation.

Microhardness as a micromechanical characteristic plays a role as a connecting link between the structure and micromechanical properties. In recent studies, microhardness was reported for different vegetable oil derivatives. Linseed oil epoxide together with epoxidized glycerol, methyl-tetrahydrophtalic anhydride as a curing agent, and 2-methylimidazole as an initiator reached a microhardness of approximately 160 MPa after 100 h of irradiation [47]. For epoxidized soybean oil, the microhardness was reported to be around 58 MPa [48]. Compared to epoxidized soybean oil, acrylated epoxidized soybean oil had a higher microhardness by more than 50%, and a slightly higher indentation elastic modulus [49]. The polyurethane derivative synthesized from rapeseed oil polyol showed a microhardness of 178 MPa [50]. However, in a dicyclopentadiene-based tetrapolymer system, linseed oil lowered the hardness [51]. Hence, it can be seen from the comparison that the obtained coatings RL_G20 and LG_G20 showed competitive microhardness or even higher values than the ones reported in the reviewed literature.

A direct relationship was observed between Vickers microhardness and Young's modulus, or tensile strength for the cured acrylate coatings with GPT. Acrylate systems showed an excellent linear relationship ($R^2 = 0.9732$) when plotting microhardness vs. E as depicted in Figure 7b. The equation is as follows: microhardness = $aE^b$ [48]. The relationship between microhardness and Young's modulus relates to materials' stiffness because both depend on the material's structure and the corresponding intra- and intermolecular interactions.

This proportional dependence has been developed for many systems, such as ultrahigh molecular weight polyethylene, copolymers of ethylene-$\alpha$-olefins, and highly oriented polyethylene [52]. It is proposed that microhardness could be used as an indicator of Young's modulus in some specimens that do not easily lend themselves to routine testing procedures [53]. A slightly poorer relationship was observed between tensile strength and microhardness ($R^2$ = 0.8317). The plywood color shifted to darker during UV exposure (Figure 7c), while coatings retained good clarity and transparency.

### 3.7. Sliding Friction

The coefficient of friction ($\mu$) between two solids is defined as F/N where F denotes the friction force and N the load or the force normal to the surface, and it is independent of the apparent area of contact [54]. To determine the performance and durability of the wood coatings after 9 h of photo-oxidation, which serves as degradation, sliding friction was measured. RL, RL_G5, and RL_G20 were selected as representative photo-oxidized coating samples for the test. Experiments were conducted under three indenter loads: 0.5, 1.5, and 3.0 N. The coefficient of friction values is displayed in Table 5 as an average of the constant plateau zone of the measurement, where the friction pair adjustment process ("run-in") and final part ("run-off") are not considered. Figure S4 in Supplementary Materials demonstrates $\mu$ vs. sliding distance or time, or laps, in which the $\mu$ at the plateau is plotted. The RL_G5 sample at load 1.5 N and 3.0 N displayed a stable $\mu$ value, increasing to a high steady-state value (0.6 < $\mu$ < 0.9) and reaching a plateau. In contrast, the sample at 0.5 N load reached the plateau gradually. When the applied load was increased from 0.5 N to 1.5 N, an increase of $\mu$ was observed. When the load was increased further up to 3N, the $\mu$ decreased for RL and RL_G20 but increased for RL_G5. A similar trend was reported with linseed oil as the metal coating. $\mu$ ranged from ~1.0 up to ~1.5 for linseed oil cured under different UV radiation intensities (100 and 300 mJ/cm$^2$), but $\mu$ decreased below 0.2 when the sample was additionally cured thermally at 80 °C after the UV irradiation [55]. Wang et al. investigated the lubricating properties of palm oil-based nanofluids. The authors reported that carbon nanotubes (CNTs) could reduce the sliding friction force due to their high strength and hardness, as CNTs will not be ground into a hard film under heavy loads [56]. Another study measured $\mu$ for epoxidized soybean oil, which was around 0.11 under dry conditions [48].

**Table 5.** Coefficient of friction values for photo-oxidized samples.

| Photo-Oxidized (9 h) Coatings | $\mu$ (in Plateau Zone) | | |
| :---: | :---: | :---: | :---: |
| | 0.5 (N) | 1.5 (N) | 3.0 (N) |
| RL | 0.24 | 0.88 | 0.79 |
| RL_G5 | 0.42 | 0.65 | 0.80 |
| RL_G20 | 0.21 | 0.47 | 0.31 |

Resistance to sliding depends on the surface morphology of the polymer, and, in many instances, this morphology is affected by sliding [57]. Optical microscopy images in Figure 8 represent the wear of coating after the test. When a hard slider is pressed into a soft polymer, the surface is extended [57]. Thereby, indentations were formed in the RL and RL_G5 coatings. Indentations for the RL and RL_G5 samples occurred when the 1.5 N load was added. However, RL_G20 remained smooth and showed only the path formed by the sliding ball. The width of the wear track increased as the applied load increased. For RL_G20, it rose from 0.242 to 0.372 and up to 0.544 mm for 0.5, 1.5, and 3.0 N, respectively. This was due to the depth of penetration of the metal ball in coating.

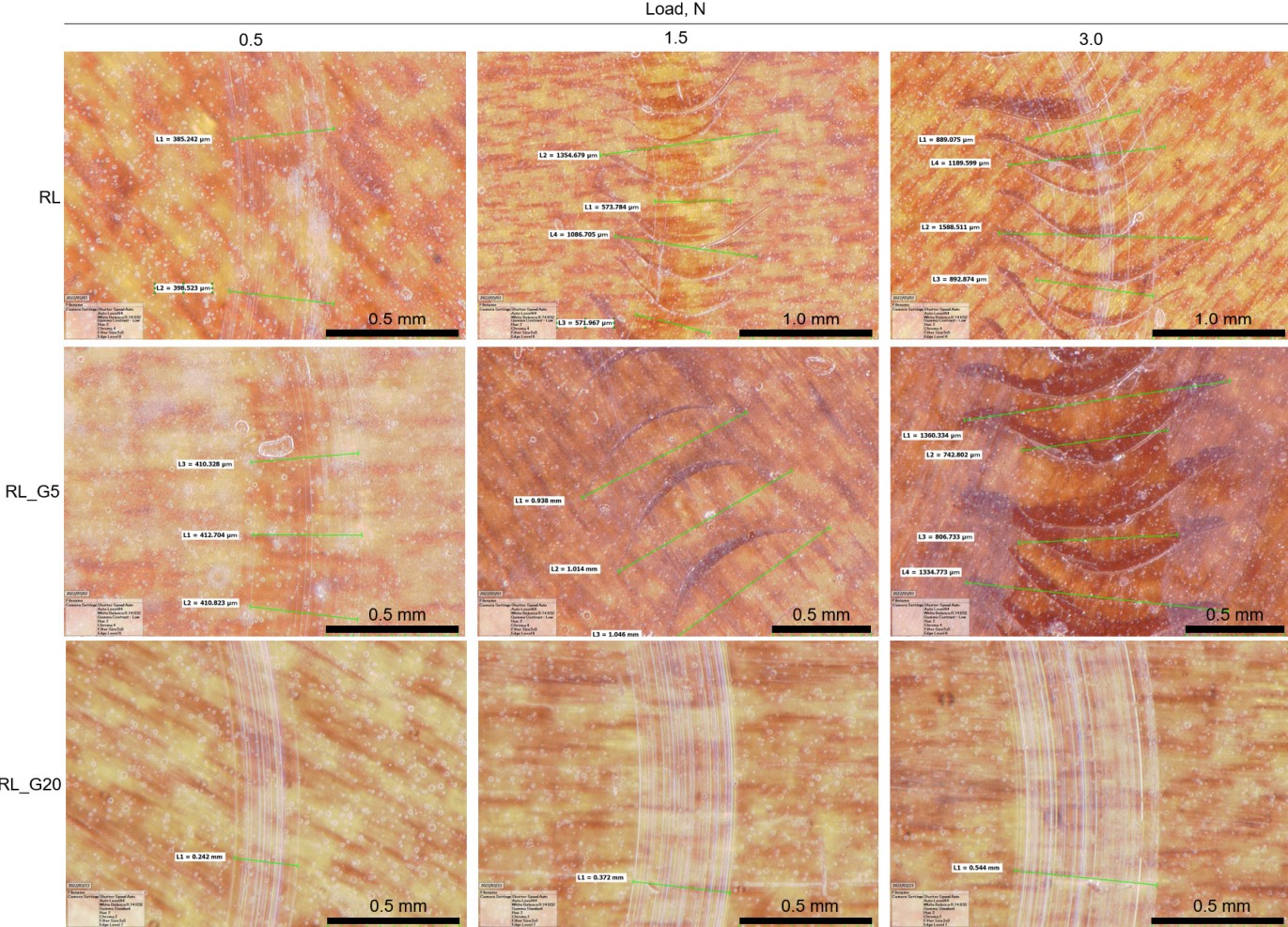

**Figure 8.** Optical microscopy photos of photo-oxidized (9 h) wood coating samples after sliding friction test at different loads.

## 4. Conclusions

The present study explored biobased resins that introduced two-phase microstructures after UV-curing. The testing was limited to potential applications for wood surfaces but was not limited to this purpose. The viscosity of AVOs decreased by an average of 550 mPa·s with the addition of a propoxylated glycerol triacrylate (GPT) monomer. The addition of the GPT monomer to acrylated vegetable oils (AVOs) resulted in two distinct effects based on the added loading: 5 or 20 wt%. The addition of 5 wt% resulted in the partial formation of the GPT network, which contributed to a pronounced two-phase separation in the microstructure and relatively large boundaries between phases. As a result, the tensile properties saw a significant decrease, and the glass transition temperature was inconsistent. Increasing GPT content yielded a much finer and well-dispersed microstructure. The addition of 20 wt% of GPT resulted in an enhanced tensile performance, dynamic mechanical properties, and increased glass transition temperature. The addition of GPT increased the storage modulus by up to five-fold, Young's modulus by up to two-fold at 20 °C, and the crosslink density by up to two-fold. Exposure to photo-oxidation enhanced the microhardness, which reached the maximum value of 250 MPa for LG_G20. The coatings remained clear and transparent after the photo-oxidation induced aging.

**Supplementary Materials:** The following supporting information can be downloaded at: https://www.mdpi.com/article/10.3390/coatings13030657/s1, Figure S1: Conversion of acrylic double bonds in coating samples; Figure S2: Viscosity and temperature relation for coating samples; Figure S3: Cross-section SEM micrographs of the fracture surface morphology; Figure S4: Photo-oxidized (9 h) coating friction coefficient dependance on applied indenter load, time, distance, and number of laps.

**Author Contributions:** Conceptualization, S.G.; methodology, S.G.; software, S.B.; validation, S.G.; formal analysis, S.B.; investigation, S.B., I.S., A.L., J.L.; resources, S.G.; writing—original draft preparation, S.B., S.G.; writing—review and editing, O.P., A.B., S.G.; visualization, S.B., S.G.; supervision, S.G.; project administration, S.G.; funding acquisition, S.G. All authors have read and agreed to the published version of the manuscript.

**Funding:** Sabine Briede was funded by the European Social Fund within the Project No 8.2.2.0/20/I/008 «Strengthening of PhD students and academic personnel of Riga Technical University and BA School of Business and Finance in the strategic fields of specialization» of the Specific Objective 8.2.2 «To Strengthen Academic Staff of Higher Education Institutions in Strategic Specialization Areas» of the Operational Programme «Growth and Employment». Sabine Briede was supported by Riga Technical University's Doctoral Grant programme.

**Institutional Review Board Statement:** Not applicable.

**Informed Consent Statement:** Not applicable.

**Data Availability Statement:** Data will be made available on request.

**Acknowledgments:** The authors wish to thank their parental institute for providing the necessary facilities to accomplish this work.

**Conflicts of Interest:** The authors declare no conflict of interest.

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
