# Peer review of "Tailored Biobased Resins from Acrylated Vegetable Oils for Application in Wood Coatings"

_coatings, doi:10.3390/coatings13030657_

Round 1
Reviewer 1 Report
The manuscript concerns the use acrylated vegetable oils for wood coatings .
The manuscript is interesting, well prepared. The planned research, and its discussion are appropriate.
1. The introduction is concise, and appropriate prepared.
2. Materials and Methods:
2.1. Please indicate the sources of the reagents used, especially vegetable oils.
2.2. What are the dimensions of coated plywood? Did cross-linking take place immediately after application of the liquid mixture? Was liquid ingress observed prior to cross-linking (referred to in section 3.5). Was the entire surface of the sample irradiated simultaneously during cross-linking?
3. Results and discussion; the discussion of the results was well conducted, without any major remarks. A large number of references are cited in the discussion.
Other minor comments are listed below:
- line 94, please correct the chemical formula, no subscripts.
- line 98; please correct photoinitiator name.
- lines 153, 166 and 197, please add references to the indicated standards.
Reviewer 2 Report
The article analysis acrylated linseed, rapeseed and grapeseed oils for wood coatings. The article is really well written, methods clearly described, the results are clearly discussed. I have no additional comments for this study. I think it is suitable for publication in Coatings MDPI.
Reviewer 3 Report
The paper deals with manufacturing resins from biological substances. These materials have primarily been proposed for wood surface treatment, accordingly, the discussed subject is suitable for the subject area covered by the journal Coatings. The paper is appealing, on the other hand, requiring much effort to comprehend from readers not directly active in the reviewed matter.
Recommendations, questions, or comments
1. Figs 3a and 3b declare the same scaling: 5000-ply magnification; however, the lengths of the referential 10 mm segments are not equal. What is correct? Comparing the structure between the films needs to use the same magnification
2. The assessment of the impairment in the system wood–coating after having performed the pull-off test needs to assess not only the surfaces on the metal discs but also the surfaces obtained after the pull-off on the substrate material (plywood). You have at your disposal a SEM microscope, use it as a tool. This is primarily important in the case of transparent materials. I am almost 100 % convinced that in all cases, there were cohesion impairments in the solid coatings. I recommend considering the work:
Kúdela, J.,Liptáková, E. 2006: Adhesion of coating materials to wood. J. Adhesion Sci. Technol., 20(8): 875-895.
3. In this case, it is also essential to consider the wood species from which the plywood, or at least, its surface veneers have been manufactured and the thickness of the solid coating applied onto the plywood.
4. On one hand, you refer low-temperature Tg values, low strength values and low values of elasticity modulus of free coating films, on the other hand, the reported hardness values are high. Have I understood correctly that these two properties were not measured under the same conditions? Is it allowable to compare in this way?
Reviewer 4 Report
In this paper, biobased UV light-curable coatings using acrylic biomass oil (rapeseed, linseed and grapeseed oil) as a substrate and biobased propoxylated glycerol triacrylate (GPT) as a diluent, which were applied to wood surfaces. The influence of the phase separation microstructure of GPT and substrate on the coating properties was highlighted. However, there are several mistakes, questions not mentioned or clearly clarified by the authors, so the manuscript should be revised before acception.
1) In line 71, Page 2, the reference [17] is followed by an extra punctuation mark.
2) In line 94, Page 2, numbers in “BF3·OEt2” are not subscripted.
3) In line 106, Page 3, the figure notes of Figure 1 are missing punctuation marks.
4) In line 206, Page 6, the initial letters of “acrylate” are not capitalized.
5) Forget spaces between numbers and letters, for example, in line 366, Page 10, in line 452, Page 12, and in line 455, Page 12 “9h”; in line 433, Page 12“3N”.
6) In line 443, Page 12, paragraphs are not indented by the first line.
7) In the Cross-section SEM micrographs of Figure 3, please supplement the original specimen SEM graphs for comparison.
8) In Figure 5, compared with the original specimen, the tensile strength of the specimen with the addition of 5% GPT decreases, while the Young's modulus increases instead. Please explain the reason for this phenomenon.
